# Deposition of Chiral Heptahelicene Molecules on Ferromagnetic Co and Fe Thin-Film Substrates

**DOI:** 10.3390/nano12193281

**Published:** 2022-09-21

**Authors:** Mohammad Reza Safari, Frank Matthes, Karl-Heinz Ernst, Daniel E. Bürgler, Claus M. Schneider

**Affiliations:** 1Peter Grünberg Institute, Electronic Properties (PGI-6), Forschungszentrum Jülich, 52428 Jülich, Germany; 2Jülich Aachen Research Alliance (JARA), Fundamentals of Future Information Technology, 52425 Jülich, Germany; 3Molecular Surface Science Group, Empa, Swiss Federal Laboratories for Materials Science and Technology, 8600 Dübendorf, Switzerland; 4Nanosurf Laboratory, Institute of Physics, The Czech Academy of Sciences, 16200 Prague, Czech Republic; 5Fakultät für Physik, Universität Duisburg-Essen, 47057 Duisburg, Germany

**Keywords:** chiral molecules, molecular deposition, ferromagnetic surfaces, adsorption geometry, STM

## Abstract

The discovery of chirality-induced spin selectivity (CISS), resulting from an interaction between the electron spin and handedness of chiral molecules, has sparked interest in surface-adsorbed chiral molecules due to potential applications in spintronics, enantioseparation, and enantioselective chemical or biological processes. We study the deposition of chiral heptahelicene by sublimation under ultra-high vacuum onto bare Cu(111), Co bilayer nanoislands on Cu(111), and Fe bilayers on W(110) by low-temperature spin-polarized scanning tunneling microscopy/spectroscopy (STM/STS). In all cases, the molecules remain intact and adsorb with the proximal phenanthrene group aligned parallel to the surface. Three degenerate in-plane orientations on Cu(111) and Co(111), reflecting substrate symmetry, and only two on Fe(110), i.e., fewer than symmetry permits, indicate a specific adsorption site for each substrate. Heptahelicene physisorbs on Cu(111) but chemisorbs on Co(111) and Fe(110) bilayers, which nevertheless remain for the sub-monolayer coverage ferromagnetic and magnetized out-of-plane. We are able to determine the handedness of individual molecules chemisorbed on Fe(110) and Co(111), as previously reported for less reactive Cu(111). The demonstrated deposition control and STM/STS imaging capabilities for heptahelicene on Co/Cu(111) and Fe/W(110) substrate systems lay the foundation for studying CISS in ultra-high vacuum and on the microscopic level of single molecules in controlled atomic configurations.

## 1. Introduction

The discovery of the chirality-induced spin selectivity (CISS) effect [1,2] as an interplay between electron spin and the handedness of chiral molecules has attracted much interest in recent years [3,4] and is expected to lead to a chirality-based quantum leap in quantum sciences [5]. The highly efficient spin-dependent electron transmission through non-conductive chiral molecules also makes the CISS effect a promising approach for future applications in spintronics, chemical sensing, enantioseparation, and enantioselective chemical and biological processes [6,7]. Several experimental approaches have been used to demonstrate the CISS effect for diverse chiral molecules and materials and have led to the following exemplary but certainly not exhaustively listed–findings:Spin-selective transmission of charge carriers through various chiral molecules ranging from long DNA strands [8] and polymers [9] to helical molecules (e.g., helicene molecules) with down to a single winding [10] yielded substantial spin polarization values, for instance, exceeding 60% for photo-emitted electrons passing through monolayers (ML) of double-stranded DNA [11] or 80% for the hole current through chiral methylbenzylammonium molecules within a layered organic–inorganic metal–halide hybrid semiconductor framework [12], both at room temperature (RT). This opens perspectives for the realization of a versatile, low-cost and energy-efficient carbon-based spintronics with reduced device size, operating at RT and not requiring ferromagnetic metals [2].The enantiospecific adsorption of various chiral molecules on ferromagnetic, perpendicularly magnetized substrates demonstrated a distinct approach to enantiomeric separations by providing a potentially generic, efficient, and cost-effective chromatographic method for enantioseparation, which does not require a specific separating column [7,13].Adsorbing a self-assembled ML of chiral molecules can switch the local magnetization of a ferromagnetic layer with perpendicular magnetic anisotropy without the need to apply an electrical current or an external magnetic field [14].

Despite the extensive experimental data on the CISS effect, its theoretical background has not yet been sufficiently clarified; thus, it is still an open question of whether there is a predictive, unifying, and experimentally tractable model of spin-dependent electron transport through a chiral molecule [5]. Most theoretical models still have some shortcomings in filling the large gap between the experimental and theoretically calculated effect size, e.g., spin polarization values or enantiospecific adsorption energies on ferromagnetic surfaces [2,15,16,17,18,19,20]. A major difficulty is the discrepancy between the experimentally studied *real systems* and the *idealized model systems* accessible to theoretical descriptions. Most experimental work involves molecular ensembles and therefore provides information averaged over many different configurations (e.g., absorption sites and geometries or electric current paths). In addition, most experiments are performed under environmental conditions that require protective coatings to prevent oxidation (e.g., Au coating of ferromagnetic substrates) or result in additional adsorbates (e.g., water due to air humidity), neither of which are accounted for in theoretical models. Therefore, single-molecule scale studies performed under well-defined vacuum conditions and in controlled geometric configurations, ideally on the atomic level, are highly desirable to gain deeper insight into the fundamentals of the CISS effect [20].

Here, we lay the foundation for such experiments by reporting the deposition of the racemic mixture of chiral heptahelicene molecules on different single-crystalline surfaces by sublimation under ultra-high vacuum (UHV, pressure lower than $10^{-7}$ Pa) conditions, where the substrate materials include the noble metal Cu and the more reactive ferromagnets Fe and Co. Heptahelicene belongs to the family of chiral helicene molecules for which the CISS effect has been demonstrated experimentally [10,21] and intensively studied theoretically [19,20,22]. We use low-temperature scanning tunneling microscopy (LT-STM) to characterize the molecular adsorption geometries and determine the handedness of individually adsorbed molecules. The latter is a prerequisite for heptahelicene on Cu(111), Co(111), or Fe(110) surfaces to be used as model systems for future experiments on the CISS effect at the single-molecule level.

## 2. Materials and Methods

Heptahelicene C_30_H_18_ ([7]H) is chiral due to seven ortho-fused benzene rings that do not fit into a plane, thus leading to a helical shape; see Figure 1. For monolayers of heptahelicene molecules on surfaces of the coinage metals Cu, Ag and Au, the CISS effect was demonstrated in photoemission experiments and resulted in 6 to 8% spin polarization [10]. The adsorption of [7]H and film growth in the ML and sub-ML regime on surfaces of the same coinage metals have been extensively studied in the context of self-assembly and crystallization [23,24,25,26,27]. However, to our knowledge, there is only one STM study on the deposition of [7]H on a ferromagnetic and more reactive surface, namely Ni(111) [28], where STM images revealed randomly oriented molecules at low coverage and a hexagonally closed-packed ML with an intermolecular distance consistent with the van der Waals radius of [7]H.

Here, we report the adsorption of [7]H molecules in the sub-ML regime on two ferromagnetic thin-film substrate systems with out-of-plane magnetization that are well-known in the magnetism community, namely Co(111) bilayer (BL) nanoislands on Cu(111) and 2 ML Fe(110) thin films on W(110). The out-of-plane magnetization leads in the STM geometry to tunneling electrons with positive or negative helicity (defined as the projection of the spin on the momentum vector), which is a prerequisite for studying the CISS effect and the interplay of spin-polarized currents with chiral molecules. The partly filled 3*d* shell of Co and Fe renders these surfaces more reactive than those of coinage metals. The adsorption of [7]H on the less reactive Cu(111) single-crystal surface between the Co nanoislands is used for comparison with the ferromagnetic surfaces as well as with previous literature. The heptahelicene molecules used in this study were synthesized at Empa, the Swiss Federal Laboratories for Materials Science and Technology, in the form of a racemic powder.

### 2.1. Sample Preparation

All experiments in this work were performed under UHV conditions in a multi-functional UHV cluster tool with a base pressure of less than $10^{-8}$ Pa, which is equipped with an LT-STM (Omicron Scienta), a preparation chamber for substrate preparation, thin-film growth and analysis, and a molecule deposition chamber. All chambers are separately pumped and can be sealed from each other.

The [7]H powder was filled without further treatment into a quartz crucible of a Knudsen cell located in the molecule deposition chamber. The sublimation of [7]H molecules sets in at a crucible temperature of about 400 K. Any contamination of the freshly prepared Co and Fe thin-film substrates due to degassing of the heater filament of the Knudsen cell during the slow and careful heat-up is detrimental to the surface magnetism and may impede spin-polarized STM imaging [29]. Therefore, in order to avoid contamination, the substrate was introduced to the molecular chamber only after reaching the sublimation temperature and was immediately exposed to the molecular vapor. The vacuum pressure during molecule sublimation did not exceed $10^{-7}$ Pa. After molecule deposition, the samples were transferred within 5 min to the STM chamber, where they were cooled to 5 K at a vacuum pressure in the mid $10^{-9}$ Pa range. We performed numerous cycles of molecular deposition, which proved the reliability and reproducibility of the deposition procedures and allowed repeated STM/STS measurements with both newly fabricated samples and modified (newly functionalized) STM tips.

In the following, we describe the procedures for preparing the three different molecule–substrate systems.

#### 2.1.1. Heptahelicene on Cu(111)

The single-crystal Cu(111) substrate was cleaned by repeated cycles of Ar^+^ sputtering and post-annealing at 1070 K according to the procedure described in [30]. After annealing, the cooling was performed with a controlled temperature gradient to promote the formation of wide atomic terraces. The sample cleanness was checked by Auger electron spectroscopy (AES) and LT-STM measurements. Heptahelicene molecules were then sublimated as described above onto the Cu(111) surface, which was at room temperature (RT).

#### 2.1.2. Heptahelicene on Co/Cu(111)

For the preparation of the [7]H/Co/Cu(111) sample, triangular Co BL nanoislands are grown on the cleaned Cu(111) crystal at RT. Co is deposited by electron-beam evaporation (triple evaporator from Scienta Omicron) with a deposition rate of 0.2 ML/min according to [31]. The result of each deposition is verified by topographic and spin-polarized STM measurements to confirm the cleanness of the surface, the density and shape of the Co nanoislands, and their well-developed magnetization perpendicular to the surface. Only when all requirements are met, heptahelicene molecules are deposited on the Co/Cu(111) substrates as described above, whereby it warms up again to RT in the molecule deposition chamber.

#### 2.1.3. Heptahelicene on Fe/W(110)

The single-crystal W(110) substrate was cleaned by repeated cycles of oxygen annealing (1000<T<2000 K) and high-temperature flashing (T>2300 K) as described in [32]. Subsequently, 1.7 ML Fe was deposited with a growth rate of 0.6 ML/min by electron-beam evaporation onto the substrate, which was at RT. Post-annealing at 570 K results in a completely filled 1st Fe ML and a partially filled 2nd Fe layer. As for the Co/Cu(111) substrate system, the Fe/W(110) film growth quality and the magnetic properties, in particular the presence of out-of-plane magnetized domains in areas covered with 2 ML Fe, were checked by topographic and spin-polarized STM measurements. The deposition of heptahelicene molecules, again with the substrate at RT, followed the procedure described above.

### 2.2. STM Measurements

All STM measurements were performed using a W tip fabricated by the electrochemical etching of a polycrystalline W wire in 5 M NaOH solution. In order to obtain spin-polarized data, we functionalized the W tip by intentionally driving it into mechanical contact with the ferromagnetic substrate remote from the region of interest and applying short voltage pulses to the tip to pick up atoms from the substrate. Lacking the possibility to apply an external magnetic field to the STM setup, we also use this procedure to modify the direction of the tip magnetization and thus the direction of its magnetic sensitivity, i.e., out-of-plane up to out-of-plane down or to in-plane sensitivity. Although this procedure is somewhat random and unpredictable, it allows us to investigate the same surface area with the tip being sensitive to different magnetization directions.

We present three types of STM data. (i) Topographic STM images were taken in constant-current mode. (ii) Differential conductivity (dI/dV) maps were taken simultaneously with constant-current topographic images (closed feedback loop). dI/dV maps represent, to a good approximation, the local density of states (LDOS) as a function of the lateral position and at the energy $eV_{\mathrm{bias}}$, where $V_{\mathrm{bias}}=0$ corresponds to the Fermi energy ($E_{F}$). (iii) Scanning tunneling spectroscopy (STS) point spectra were acquired at a fixed lateral and vertical tip position (open feedback loop) by sweeping $V_{\mathrm{bias}}$. dI/dV point spectra represent, to a good approximation, the LDOS as a function of the energy $eV_{\mathrm{bias}}$ at a fixed tip position. The dI/dV signal for the conductivity maps and point spectra was obtained by superimposing $V_{\mathrm{bias}}$ with a small sinusoidal modulation (rms amplitude $V_{\mathrm{mod}}=10$ or 20 mV; frequency $f_{\mathrm{mod}}=875$ or 752 Hz) and detecting the resulting modulation of the tunneling current using a lock-in amplifier. In the present case, we employ topographic images (STM data type (i)) to address the quality of the substrates as well as the integrity, adsorption geometry, adsorption position and chirality of the deposited molecules. dI/dV maps (STM data type (ii)) obtained with magnetic tips reveal the ferromagnetism and magnetic domain structure of the Co and Fe substrates. dI/dV point spectra (STM data type (iii)) measured on adsorbed molecules show the different impact of the Cu, Co and Fe substrates on the LDOS of the physisorbed and chemisorbed [7]H molecules, respectively.

## 3. Results

### 3.1. Heptahelicene on Cu(111)

Figure 2a shows a constant-current topographic image of a sub-ML coverage of the Cu(111) surface with [7]H molecules. A Cu(111) monatomic step is visible on the right side of the image, which is densely decorated with molecules. The STM data reveal a clean deposition of intact [7]H molecules, since fragments of molecules and other adsorbates are barely visible. The molecules mostly form dimers with only a few trimers and monomers present as reported by Ernst et al. [25], who investigated the structure and binding of the dimers in detail. Both the dense step edge decoration and the almost complete dimer formation indicate the high surface mobility of the molecules at RT. Note that the equilibrium configuration of the molecules on the surface is reached during and immediately after the deposition at RT, and the STM images show a frozen state after cooling the sample to 5 K.

Figure 2b shows a submolecularly resolved image of three dimers. Each molecule appears as a radially asymmetric gray halo with a bright spot that is offset from the center of the halo. The cross-section along the red line in Figure 2b, displayed in Figure 2d, reveals an apparent height of the bright spots of 240 pm, which indicates an adsorption geometry of the [7]H molecule with its helix axis perpendicular to the surface. This is in agreement with an X-ray photoelectron diffraction study of Cu(111) covered with 1/3 ML [7]H [23].

These data agree very well with a previous report [25] on the deposition of a racemic mixture of [7]H molecules on Cu(111). The authors related the off-centered maximum and the radially asymmetric, hence helical, gray halo of the apparent height profile to the handedness of the [7]H molecules and found that the dimers are heterochiral; i.e., each dimer consists of one (*M*)-[7]H and one (*P*)-[7]H molecule. Similar to Ernst et al. [25], we apply a Gaussian high-pass filter to the topographic STM image in Figure 2b and obtain a better visualization of the molecules’ handedness in Figure 2c, where the handedness can be directly determined, as indicated by the dashed blue and red circular arrows. Technical information about the Gaussian high-pass filter is detailed in Appendix A. In the following, we will use dashed blue and red circular arrows to label right-handed (*P*)-[7]H and left-handed (*M*)-[7]H molecules, respectively. The arrows are drawn to start adjacent to the proximal phenanthrene group close to the surface and with the arrowheads adjacent to the distal end (see scheme in Figure 2e).

In order to improve the statistics, we applied the Gaussian high-pass filter to high-resolution data of a larger area marked by a red square in Figure 2a. The result is shown in Figure 2f. For molecules not marked with a dashed arrow, the assignment of absorption orientation and handedness was not possible. This can be caused by adsorption on a defect of the substrate or by aggregation with more than one other molecule. Evidently, the heterochiral nature of the dimers is clearly confirmed. In addition, we find only three rotational orientations of the adsorbed molecules, which occur with approximately equal probability (see Table 1): The bright spot is offset from the halo center in the $\left[\bar{1}\;1\;0\right]$, $\left[1\;0\;\bar{1}\right]$, or $\left[0\;\bar{1}\;1\right]$ direction of the Cu(111) surface. The observation of three distinct adsorption orientations is in agreement with the above-mentioned photoelectron diffraction study of [7]H on Cu(111) [23]. In Appendix B, we present a larger data set for which we have performed a so-called mosaic measurement that includes a set of high-resolution topography images. By merging these images, we have data from a more extended area of the sample surface with sufficient resolution for analysis. The evaluation of 593 [7]H molecules again yields an even distribution on the three adsorption orientations specified in Figure 2e (see 1st row in Table 1).

In summary, the [7]H molecules adsorb on Cu(111) in a configuration, in which the molecule spirals away from the surface and forms in the sub-ML regime mostly heterochiral dimers. Within dimers, the [7]H molecules adsorb at a well-defined adsorption site in three orientations with respect to the substrate, which result from the three-fold symmetry of the (111) surface of the face-centered cubic (fcc) Cu crystal, making them energetically degenerate and evenly occupied.

### 3.2. Heptahelicene on Bilayer Co Nanoislands

The formation, atomic structure, electronic and magnetic properties of Co BL nanoislands on Cu(111) have been extensively studied and are well established [31,33,34,35,36,37]. Figure 3a shows the result of depositing a sub-ML amount of Co on Cu(111) at RT. Triangular islands are formed with a typical size of 10 nm and an apparent height of 370 pm, corresponding to an island height of 2 ML (see blue cross-section in Figure 3b taken along the blue line in Figure 3a). The two different island orientations, which are rotated by 60° with respect to each other, result because the first layer’s Co atoms can adsorb on Cu(111) to fcc or hexagonal-closed-packed (hcp) sites, leading to so-called “unfaulted” and “faulted” islands, respectively [31,35]. On both island types, the second-layer Co atoms can also adsorb to either fcc or hcp sites. In all four cases, the atomic structure of the island surfaces is three-fold symmetric. Islands with throughout fcc or hcp stacking have the same symmetry-equivalent in-plane directions as the Cu(111) surface below, while on islands with mixed stacking, all in-plane directions are inverted.

Figure 3c shows the differential conductivity (dI/dV) map at −600 mV recorded simultaneously with the topographic image in Figure 3a and using a Co-functionalized W tip. The Co nanoislands can be divided into two groups with distinctly different differential conductivity, i.e., LDOS at −600 meV. Figure 3d shows another dI/dV map at −600 mV taken in the same sample area as Figure 3a,c but after intentional modification of the Co-functionalized W tip on a remote Co nanoisland as described in Section 2.2. Evidently, the differential conductivity of all islands has changed from low to high or vice versa. An exception is the nanoisland in the red frame in Figure 3a, which, however, is 3 instead of 2 ML high (see red line profile in Figure 3b) and is therefore not of further interest here, since Co films on Cu(111) thicker than 2 ML are in-plane magnetized [38]. Consistent with previous reports [33,34,36,37,39], the different dI/dV contrast of the islands and its inversion in Figure 3c and d arise from spin-polarized tunneling between the out-of-plane magnetized Co nanoislands [33] and the Co-functionalized W tip. The islands appear light or dark in the dI/dV maps if the magnetization of the tip is parallel or antiparallel to the island magnetization, respectively. The contrast reversal from Figure 3c,d is due to a reversal of the tip magnetization caused by the intentional modification of the tip. Thus, the Co/Cu substrate system provides oppositely out-of-plane magnetized, single-domain Co nanoislands with bare Cu(111) regions in between within an area of less than 200 × 200 nm^2^.

The result of depositing a sub-ML amount of [7]H molecules on the Co/Cu substrate system at RT is shown in Figure 4. The constant-current topographic image in Figure 4a shows that all molecules are adsorbed on the Co nanoislands, and not a single molecule is found on the Cu surface. The apparent height (about 250 pm) and shape of the molecules are similar to those in Figure 2, implying an adsorption geometry on Co(111) in which the lower phenanthrene group is also aligned parallel to the surface. The high surface mobility of the [7]H molecules on the Cu surface at RT allows them to diffuse to Co nanoislands, where they accumulate in the thermal equilibrium due to the much stronger binding to the more reactive Co [40]. The numerous adsorption along the island edges is due to the enhanced interaction at step edges and with the Co rim states [34]. The dI/dV map in Figure 4b recorded simultaneously with Figure 4a reveals the magnetic contrast of the Co nanoislands in the presence of [7]H molecules. Note that in contrast to Figure 3c,d, we used a color scale with multiple color transitions to visualize both the different LDOS due to the oppositely out-of-plane magnetized islands (dark and red) and the enhanced LDOS above the [7]H molecules and the rim state (green-blue). These data indicates a clean molecule sublimation process that does not affect the magnetism of the Co nanoislands.

Figure 4c shows a high-resolution topographic image of the area marked in red in Figure 4a, to which we have applied the Gaussian high-pass filter. Similar to the case of [7]H on Cu(111) in Section 3.1, the filtered data show submolecular features that allow determining the molecules’ handedness and adsorption orientation, as indicated by the blue and red circular arrows. We observe only three adsorption orientations, which are exactly opposite to those found for [7]H on Cu(111): $-[\bar{1}\;1\;0]$, $-[1\;0\;\bar{1}]$, and $-[0\;\bar{1}\;1]$, where the Miller indices always refer to the lattice of the Cu substrate. We performed the same analysis for all Co nanoislands in Figure 4a and found for each island, that there are only three absorption orientations. For some islands, they coincide with those of [7]H on Cu(111) (i.e., $\left[\bar{1}\;1\;0\right]$, $\left[1\;0\;\bar{1}\right]$, and $\left[0\;\bar{1}\;1\right]$, see Figure 2) and for the others with those in Figure 4c (i.e., $-[\bar{1}\;1\;0]$, $-[1\;0\;\bar{1}]$, and $-[0\;\bar{1}\;1]$). Both groups include both faulted and unfaulted islands. These observations, in combination with the different stacking sequences of the four island types, lead to the conclusion that there are only three adsorption orientations of the [7]H molecule with respect to the atomic lattice of the Co BL nanoislands. Two exactly opposite adsorption orientations with respect to the Cu substrate (e.g., $\left[\bar{1}\;1\;0\right]$ and $-[\bar{1}\;1\;0]$) that we find on different islands correspond to the same orientation with respect to the respective Co lattice. In the 3rd row of Table 1, we summarize the analysis of 152 molecules on all islands in Figure 4a and report the totals for the three orientations with respect to the Co lattice, which we have labeled $\pm [\bar{1}\;1\;0]$, $\pm [1\;0\;\bar{1}]$, and $\pm [0\;\bar{1}\;1]$. Similar to adsorption on Cu(111), we find a uniform distribution within the limits of statistics.

In summary, the [7]H molecules adsorb on Co nanoislands on Cu(111) in a configuration with the helix axis perpendicular to the surface and occur in the sub-ML regime as monomers with a preference to occupy the island edges. The molecules adsorb at a well-defined adsorption site in three in-plane orientations with respect to the Co lattice that coincide with those found in Section 3.1 for [7]H on Cu(111). The uniform distribution of the three orientations indicates their energetic degeneracy resulting from the three-fold symmetry of the atomic lattice of the Co nanoislands. Sub-ML coverage with [7]H molecules does not affect the magnetism of the Co nanoislands, which remain in a single-domain state and magnetized out-of-plane.

### 3.3. Heptahelicene on Ultra-Thin Fe Films on W(110)

Ultra-thin, epitaxially grown Fe films on W(110) also represent a well-established magnetic substrate system that has been studied in detail [41,42,43,44,45]. The deposition of slightly less than 2 ML Fe on W(110) followed by post-annealing results in sample regions covered with 1 or 2 ML Fe. The BL regions often extend from step edges of the W crystal across the lower-lying terraces. This can be seen in Figure 5a, which presents a constant-current topographic image of a 1.7 ML thick Fe film on W(110) measured with a Fe-functionalized W tip. Regions covered by 1 or 2 ML Fe are marked with ‘ML’ and ‘BL’, respectively. The dashed lines signify monatomic step edges of the W(110) crystal surface underneath. Bright lines occurring exclusively on BL regions and running along the [001] direction (red arrows) are due to characteristic dislocation lines. They arise from the relatively big lattice mismatch of 9.4% between Fe and W [41]. The simultaneously recorded dI/dV map at 50 mV in Figure 5b probes the spin-polarized $d_{z^{2}}$ orbitals and reveals a rich magnetic domain structure, similar to previous reports [29,42,43,44]. The dark brown-bluish and bright yellow colors in the BL regions represent oppositely magnetized out-of-plane domains separated by mostly horizontal dark blue domain walls. The in-plane magnetized ML regions [42,45] and the dislocation lines also appear in dark blue. Detailed analysis of domain wall profiles, which is beyond the scope of this work, reveals contrasts due to spin-polarized tunneling from the tip with an out-of-plane magnetization component as well as due to tunneling anisotropic magnetoresistance [46,47]. Thus, the Fe/W substrate system provides in the BL regions oppositely out-of-plane magnetized domains that extend across dislocation lines. Unperturbed areas of the domains suitable for molecular adsorption studies range in size from about 20 to 40 nm and are bounded by either 8 nm wide domain walls or dislocation lines.

Figure 6a shows the successful deposition of [7]H molecules on the Fe surface. Bright spots visible on both ML and BL regions are single [7]H molecules. In BL regions, [7]H preferentially adsorbs on the dislocation lines that are known to be more reactive than the unperturbed Fe BL surface between the dislocation lines [29,48]. This indicates that the molecules are mobile in the BL regions immediately after the adsorption at RT. However, in ML regions, the [7]H molecules are more randomly distributed and do not preferentially adsorb at step edges, which is probably due to the lower surface mobility. The simultaneously measured dI/dV map in Figure 6b displays the typical domain structure of pristine, out-of-plane magnetized BL regions with domain walls running along $\left[1\bar{1}0\right]$, even in the presence of [7]H molecules. This observation proves a clean molecule deposition process, since any absorbed contamination on the highly reactive surface of ultra-thin Fe film would impair the magnetic properties [29].

Figure 6c shows a high-resolution topographic image of [7]H molecules on an unperturbed Fe BL region after applying Gaussian high-pass filtering. The submolecular resolution allows determining the handedness and the absorption orientation of the molecules, similar to [7]H on Cu(111) (Section 3.1) or Co BL nanoislands (Section 3.2). Examples are shown in the inset. Only two adsorption orientations are observed, and the position of the molecule’s highest apparent height is located in the $\left[\bar{1}10\right]$ or [001] direction from the molecule center. Interestingly and in contrast to [7]H on Cu(111) (Figure 2f) or Co BL nanoislands (Figure 4c), we observe a correlation between absorption orientation and handedness: left-handed molecules (red circular arrow) absorb in the $\left[\bar{1}10\right]$ orientation and right-hand molecules absorb in the [001] orientation, respectively. The analysis of 100 molecules in Figure 6c yields 43 molecules with $\left[\bar{1}10\right]$ orientation and 44 molecules with [001] orientation (see 4th row of Table 1), while the orientations of 13 molecules remained unknown.

In summary, [7]H molecules adsorb on Fe BL regions on W(110) in a configuration with the helix axis perpendicular to the surface and occur in the sub-ML regime as monomers with a strong preference to occupy dislocation lines. In unperturbed areas between dislocation lines, the molecules adsorb at a well-defined site in only two in-plane orientations with respect to the non-uniformly strained Fe lattice. Each of the two orientations is occupied by molecules with a particular handedness, and their overall equal abundance reflects the racemicity of the molecular sample. The symmetry of an fcc (110) surface suggests a larger number of possible absorption orientations than actually observed, indicating a chirality dependence of the adsorption energy. Sub-ML coverage with [7]H molecules does not affect the magnetism of the Fe BL regions, which remain out-of-plane magnetized and exhibit a domain structure similar to pristine Fe BL regions.

### 3.4. Spectroscopy of Molecules on Different Substrates

To gain more information about the interaction of the [7]H molecules with the different substrates, we recorded by STS local dI/dV spectra over [7]H molecules adsorbed on Cu(111), Co/Cu(111), and Fe/W(110).

In general, the results of STS measurements are highly dependent on the characteristics of the tip, and different tip conditions may cause different spectral features. The wider range of bias voltages applied in STS measurements makes structural and electronic changes of the tip more likely than during STM imaging with fixed $V_{\mathrm{bias}}$. Therefore, we follow a procedure that includes reference measurements to ensure that the tip has not changed during an STS measurement. We start by recording an STS spectrum at a position above the bare substrate (labeled ‘substrate before’) and compare the result with the STS spectra from the literature, i.e., [49] for Cu(111) and Co BL islands on Cu(111) and [50] for Fe BL regions on W(110). If the spectra are similar, we perform STS on nearby adsorbed [7]H molecules. Finally, we return to the initial position above the bare substrate and record another spectrum (labeled ‘substrate after’) to ensure that the tip condition has not been changed during the sequence of the measurements. We also repeat each STS measurement five times to improve the signal-to-noise ratio of the STS spectra.

Figure 7a presents STS curves of molecules on the Cu(111) surface. The relatively flat STS spectra of the bare Cu substrate (red and black) result from delocalized *s* and *p* states near $E_{F}$. The spectrum of the molecule (blue) also exhibits a similarly flat, featureless curve within ±1 eV around $E_{F}$. The inset shows the spectra in a wider energy range from −2.5 to 2.5 eV. The shoulder at about 1.5 eV and the signal increases at ±2.5 eV arise from the molecular states, since the substrate contribution is structureless at these energies. The approximately 2.5 to 3.0 eV wide gap between the onsets of the highest occupied molecular orbital (HOMO) at −2.0 eV and the lowest unoccupied molecular orbital (LUMO) at 0.5–1.0 eV is in good agreement with the calculated HOMO–LUMO gap of the [7]H molecule in the gas phase [10]. This observation supports our expectation that [7]H physisorbes on the rather inert Cu(111) surface, which causes only minor changes in the molecular orbitals of the free molecule. The low adsorption energy of the physisorbed molecules renders the monomer unstable and causes dimer formation even at low coverages, which is in agreement with the observation in Figure 2.

STS spectra of the heptahelicene molecules on Co BL nanoislands are shown in Figure 7b. The red and black spectra taken above the bare Co island are dominated by the *d*-like Co surface state at approximately −0.3 eV. The spectrum taken above the molecule (blue) does not exhibit similarly sharp states but rather a broad, gapless intensity distribution that indicates the hybridization of the molecular orbitals with the Co(111) substrate, which is more reactive compared to Cu(111) [40]. Hence, [7]H chemisorbs on Co(111) and occurs as a monomer (Figure 4).

A qualitatively similar situation is seen in Figure 7c for [7]H adsorbed on Fe BL regions on W(110). The red and black spectra of the bare Fe BL region exhibit the expected sharp peaks at −0.08 and +0.75 eV, which are due to the characteristic $d_{z^{2}}$ orbitals of BL Fe/W(110). The spectrum taken above the molecule (blue) has neither a gap nor sharp features. The broad peak at approximately 0.85 eV indicates the strong hybridization of molecular orbitals with Fe $d_{z^{2}}$ orbitals of the substrate, similar to the increasing intensity in the [7]H/Co/Cu(111) spectrum in Figure 7b below −0.25 eV. The strong modification of the molecular LDOS upon adsorption reveals that [7]H monomers also chemisorb on BL Fe on W(110). The expected larger binding energy on Fe(110) compared to Co(111) [40] cannot be inferred from the spectra in Figure 7b,c.

## 4. Discussion

We have demonstrated the deposition of heptahelicene molecules by sublimation under UHV conditions onto structurally and magnetically well-defined substrates, namely Co(111) BL nanoislands on Cu(111), Fe(110) BL films on W(110) and, for reference, Cu(111). On all three substrates, the molecules adsorb in the sub-ML regime in a configuration with the helical axis perpendicular to the surface plane. In this configuration, the electron propagation direction in transport experiments using an STM is colinear to the helical axis of the chiral molecule, which is required for observing the CISS effect. Statistic analyses of the in-plane adsorption orientations evidence that the [7]H molecules adsorb at unique and well-defined sites of the substrate lattices. On Cu(111) and Co(111), there are three equally occupied and therefore degenerate adsorption orientations with intermediate angles of 120°, corresponding to a single adsorption configuration that occurs in three symmetry-equivalent orientations with respect to the atomic Cu or Co substrate lattices. For Fe(110), only two in-plane orientations of [7]H are observed, each occupied by only one particular enantiomer. This correlation between the absorption orientation and handedness of the molecule as well as the lower number of orientations than expected based on the symmetry of the substrate lattice suggest a chirality dependence of the adsorption of [7]H on Fe(110), but this is beyond the scope of the present work. The observation of unique adsorption sites results from the sufficiently high surface mobility of the [7]H molecules after deposition at RT not only for physisorption on the noble metal Cu but also for chemisorption on the more reactive ferromagnets Fe and Co. The preferred adsorption at edges of Co nanoislands and dislocation lines on Fe confirm the high surface mobility, which allows the molecules to reach energetically favorable sites. This knowledge is of utmost importance for the correct atomic modeling of the adsorption configurations in theoretical simulations of the CISS effect.

High-resolution topographic STM images enable determining of the molecular handedness in a similar (and here reproduced) manner as previously reported for [7]H physisorbed on Cu(111) [25], but it is now extended to [7]H chemisorbed on ferromagnetic Co(111) and Fe(110) surfaces. The discrimination of discretely adsorbed enantiomers opens the door for the observation and investigation of the CISS effect at the single-molecule level. The ferromagnetic substrate systems employed here offer two advantages for STM/STS experiments on the CISS effect: First, the out-of-plane magnetization provides electrons with the spins aligned parallel or antiparallel to the tunneling direction, which maximizes the CISS-derived interaction that can be phenomenologically described as the scalar product of the molecule and electron helicities. Second, their well-known magnetic domain structure can be used to confirm and calibrate the spin-resolving capability of STM tips in SP-STM/STS measurements, without which a contrast difference (e.g., between two adsorbed molecules) cannot be unambiguously assigned to a magnetic origin. To this end, we have confirmed that magnetic contrast revealing the domain structure can be achieved on both Co(111) nanoislands and Fe BL regions, which is also in the presence of sub-ML coverage with [7]H molecules.

In conclusion, the presented capabilities for the deposition control and STM/STS imaging of [7]H molecules on Co/Cu(111) and Fe/W(110) substrate systems lay the foundation for studying the CISS effect under well-defined vacuum conditions and on the microscopic level of single molecules in controlled atomic configurations. In particular, enantiospecific adsorption in ferromagnetic surfaces and chirality-induced spin polarization can be addressed by SP-STM/STM. The microscopic structural, electronic, and magnetic characterization of the molecule–substrate systems will make the results readily accessible to theoretical analysis and modeling and shed light on the microscopic origin of the CISS effect.

## Figures and Tables

**Figure 1 nanomaterials-12-03281-f001:**
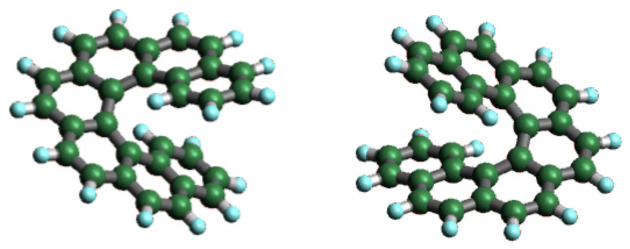
Schematic representation of the helical left-handed (*M*)-[7]H and right-handed (*P*)-[7]H.

**Figure 2 nanomaterials-12-03281-f002:**
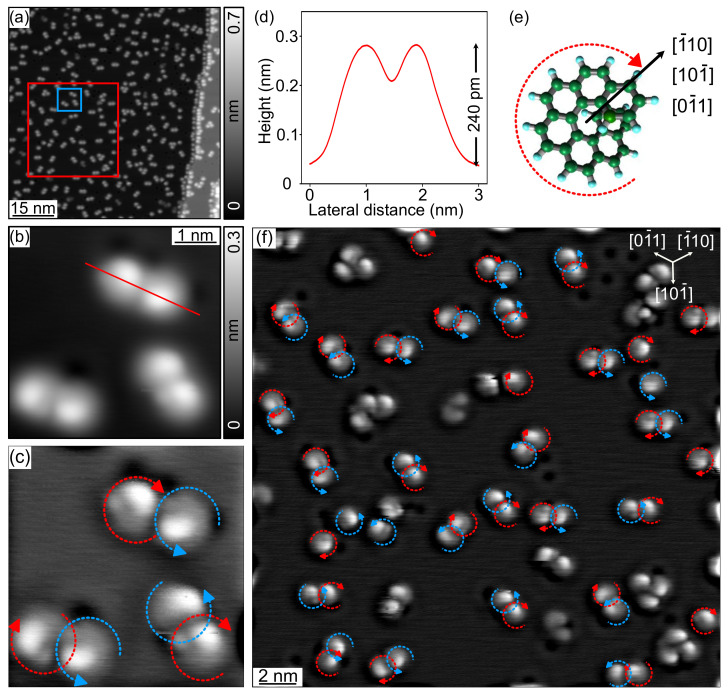
(**a**) Constant-current topographic image of [7]H molecules on the Cu(111) surface ($V_{\mathrm{bias}}=500$ mV, $I_{t}=100$ pA). (**b**) Zoom view of the blue marked area in (**a**) showing [7]H molecules with submolecular resolution ($V_{\mathrm{bias}}=500$ mV, $I_{t}=200$ pA). (**c**) High-pass filtered data of (**b**). (**d**) Cross-section (averaged over a line width of 10 pixels) along the red line in (**b**) showing the apparent height profile of a (*M*)-[7]H–(*P*)-[7]H dimer. (**e**) Schematic representation of the position of a [7]H molecule’s topmost C hexagon and the dashed arrow marking the handedness and in-plane adsorption orientation (black arrow), which points on the Cu(111) surface along the $\left[\bar{1}\;1\;0\right]$, $\left[1\;0\;\bar{1}\right]$, or $\left[0\;\bar{1}\;1\right]$ direction. (**f**) High-resolution image of the red marked area in (**a**) after applying the Gaussian high-pass filter ($V_{\mathrm{bias}}=500$ mV, $I_{t}=100$ pA). Dashed blue and red circular arrows mark right-handed (*P*)-[7]H and left-handed (*M*)-[7]H molecules, respectively. All STM data were measured at 5 K with a W tip.

**Figure 3 nanomaterials-12-03281-f003:**
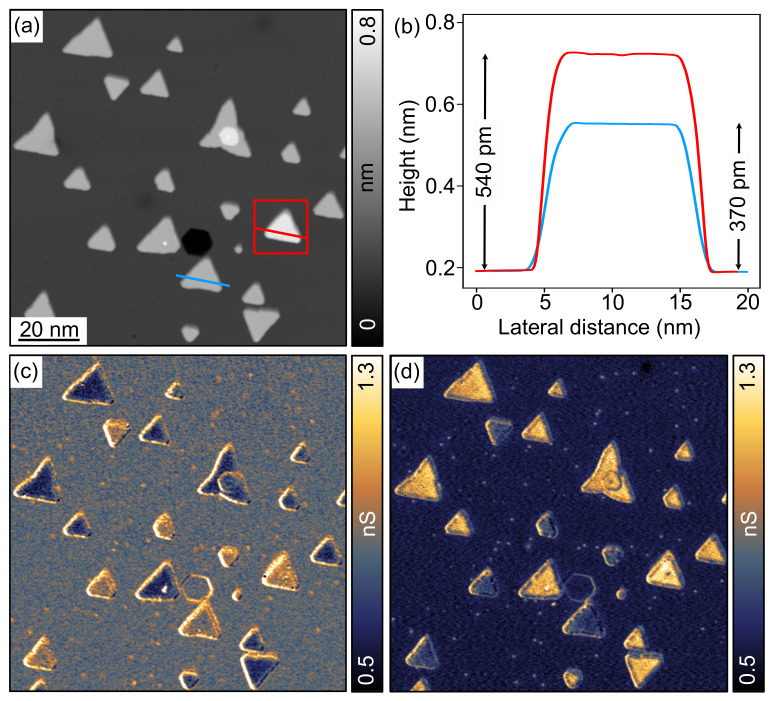
(**a**) Constant-current topographic image of Co nanoislands on Cu(111). (**b**) Line profiles (averaged over a line width of 10 pixels) along the blue and red lines in (**a**). (**c**) dI/dV map at −600 mV measured simultaneously with the topographic image in (**a**). (**d**) dI/dV map at −600 mV of the same area as in (**a**,**c**) after reversing the tip magnetization. All data are measured with $V_{\mathrm{bias}}=-600$ mV, $I_{t}=1$ nA, $V_{\mathrm{mod}}=10$ mV, $f_{\mathrm{mod}}=875$ Hz and at 5 K using a Co-functionalized W tip.

**Figure 4 nanomaterials-12-03281-f004:**
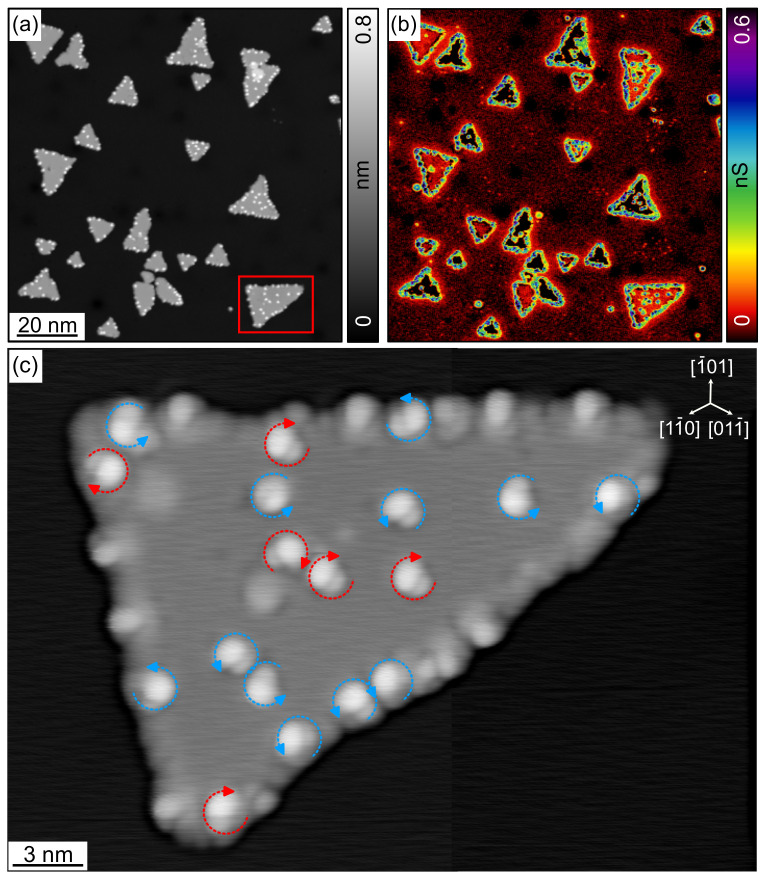
(**a**) Constant-current topographic image of [7]H molecules on Co nanoislands on Cu(111). (**b**) Simultaneously with (**a**) measured dI/dV map at −600 mV ($V_{\mathrm{bias}}=$-600 mV, $I_{t}=900$ pA, $V_{\mathrm{mod}}=20$ mV, $f_{\mathrm{mod}}=752$ Hz, 5 K, Co-functionalized W tip). (**c**) High-resolution topographic image of the red marked area in (**a**) after subsequent Gaussian high-pass filtering. Dashed blue and red circular arrows indicate the handedness and in-plane adsorption orientation of the [7]H molecules ($V_{\mathrm{bias}}=1000$ mV, $I_{t}=100$ pA, 5 K).

**Figure 5 nanomaterials-12-03281-f005:**
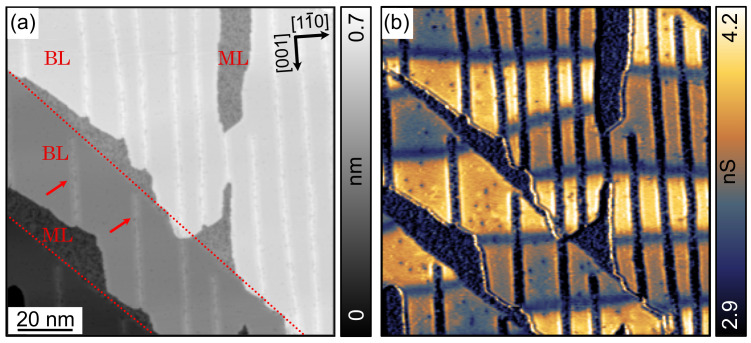
(**a**) Constant-current topographic image of a 1.7 ML thick Fe film on W(110). ML and BL regions are labeled, and arrows point to dislocation lines. (**b**) Simultaneously recorded dI/dV map ($V_{\mathrm{bias}}=50$ mV, $I_{t}=300$ pA, $V_{\mathrm{mod}}=10$ mV, $f_{\mathrm{mod}}=875$ Hz, 5 K, Fe-functionalized W tip).

**Figure 6 nanomaterials-12-03281-f006:**
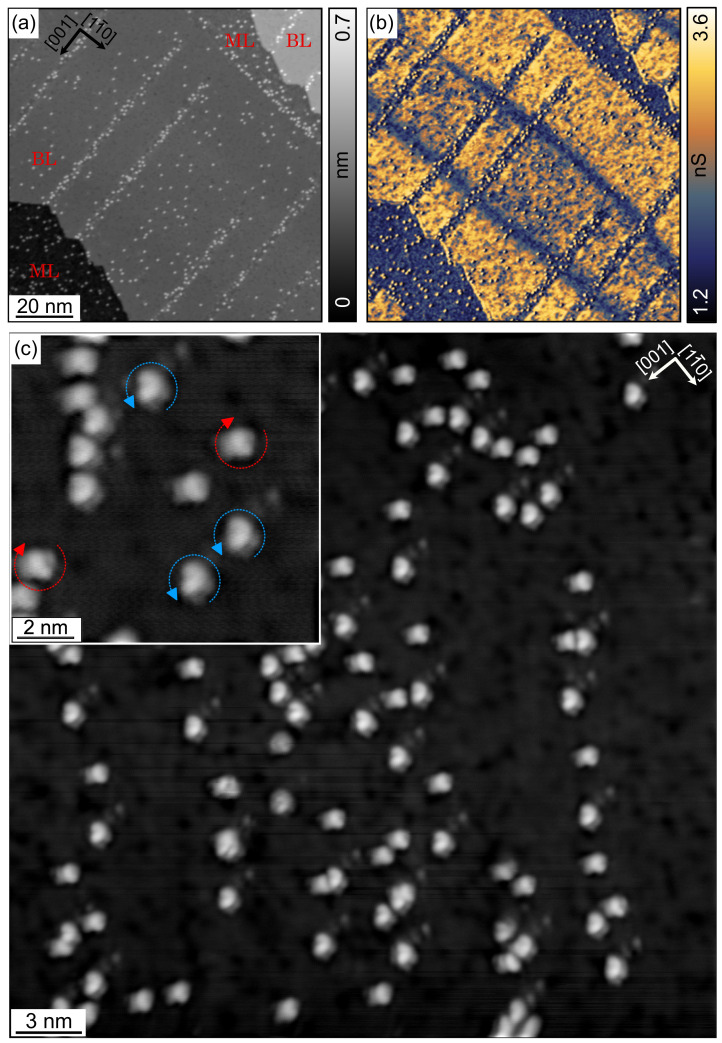
(**a**) Constant-current topographic image of [7]H molecules on Fe/W(110). (**b**) The dI/dV map measured simultaneously with (**a**) reveals that out-of-plane magnetized domains occur in BL regions also in the presence of [7]H molecules, similar to the pristine case in Figure 5. (**c**) High-resolution topographic image of [7]H molecules on an unperturbed Fe BL region after applying Gaussian high-pass filtering. From these data, the molecular handedness and adsorption orientation can be determined, as exemplified by dashed blue and red circular arrows in the inset. Data have been measured at $V_{\mathrm{bias}}=50$ mV, $I_{t}=300$ pA, $V_{\mathrm{mod}}=10$ mV, $f_{\mathrm{mod}}=875$ Hz, 5 K, and by means of a Fe-functionalized W tip for (**a**,**b**) and a non-magnetic W tip for (**c**).

**Figure 7 nanomaterials-12-03281-f007:**
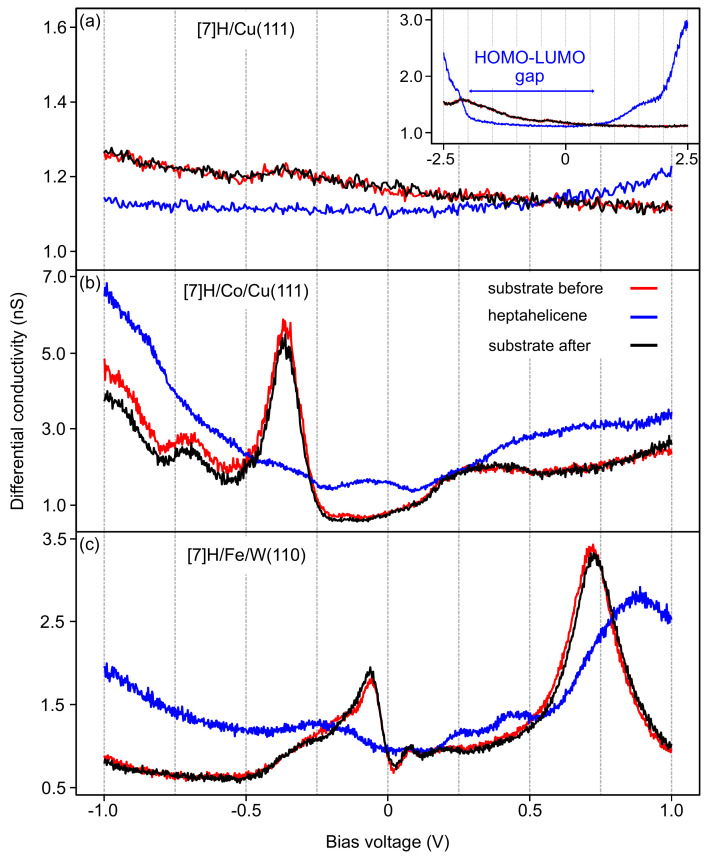
STS spectra of [7]H molecules on (**a**) Cu(111), (**b**) Co BL nanoislands on Cu(111), and (**c**) Fe BL regions on W(110). The inset in (**a**) shows a wider bias voltage range. Red and black curves are reference spectra of the bare substrate taken before and after acquiring the blue spectrum over an adsorbed [7]H molecule. The tip position has been stabilized during the STS measurements at $V_{\mathrm{bias}}=1$ V and $I_{t}=1$ nA. $V_{\mathrm{mod}}=20$ mV [10 mV for [7]H/Cu(111)], $f_{\mathrm{mod}}=752$ Hz [875 Hz for [7]H/Cu(111)], 5 K.

**Table 1 nanomaterials-12-03281-t001:** Statistics of the [7]H adsorption orientations on Cu(111), Co/Cu(111), and Fe/W(110). On Cu(111), [7]H absorbs in three equally populated and thus degenerate orientations. Co BL nanoislands on Cu(111) occur in four different stacking sequences that can be assigned to two orientations of the Co lattice with respect to the Cu substrate. The plus sign in, for example, $\pm [\bar{1}10]$ refers to the Co lattice aligned to the Cu lattice, and the minus sign refers to the inverted Co lattice. Hence, [7]H adsorbs uniformly distributed in three orientations with respect to the atomic lattice of the Co BL nanoislands. On Fe/W(110), [7]H adsorbs with equal probability in only two orientations. The error margins are the statistical errors of the counts.

Substrate (Data)	Analyzed Molecules	Orientation 1	Orientation 2	Orientation 3
Cu(111) (Figure 2f)	56	$\left[\bar{1}10\right]$: 20 ± 4	$\left[10\bar{1}\right]$: 21 ± 5	$\left[0\bar{1}1\right]$: 15 ± 4
Cu(111) (Figure A2)	593	$\left[\bar{1}10\right]$: 205 ± 14	$\left[10\bar{1}\right]$: 202 ± 14	$\left[0\bar{1}1\right]$: 186 ± 14
Co/Cu(111) (Figure 4a)	152	$\pm [1\bar{1}0]$: 50 ± 7	$\pm [\bar{1}01]$: 45 ± 7	$\pm [01\bar{1}]$: 57 ± 8
Fe/W(110) (Figure 6e)	87	[001]: 44 ± 7	$\left[\bar{1}10\right]$: 43 ± 7	-

## Data Availability

We can provide original data on request.

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
