# Peer review of "Deposition of Chiral Heptahelicene Molecules on Ferromagnetic Co and Fe Thin-Film Substrates"

_nanomaterials, 2022, doi:10.3390/nano12193281_

Round 1

Reviewer 1 Report

In this manuscript, the authors have reported their study on the deposition of chiral heptahelicene by sublimation under ultra-high vacuum onto bare Cu(111), Co bilayer nanoislands on Cu(111), and Fe bilayers on  W(110) by low-temperature spin-polarized scanning tunneling microscopy/spectroscopy. In all cases, the molecules remain intact and adsorb with the proximal phenanthrene group aligned parallel to the surface. Overall the manuscript is well organized and the language is global good. In my opinion, their work laid the foundation for studying CISS in ultra-high vacuum and on the microscopic level of single molecules in controlled atomic configurations. Thus, I recommend this manuscript published on nanomaterials as it is.  

Reviewer 2 Report

The authors present an interesting work on the interaction between electron spin and handedness of chiral molecules in surface-adsorbed chiral molecules. I have several questions need to be addressed to clarify my doubt.

1. The author should provide more information on the important of CISS effect to highlight the motivation of the work

2.  In the experimental work, author should provide the quantitative vacuum values to edge other reader to follow.

3. In the STM measurement, the author should explain why you do three different measurments. In the The dI/dV signal for the conductivity maps and point spectra, what is the important of the chosen frequencies of the envelop signal? Why do you select fmod = 875 of 752 Hz?

4. Do you develop the Gaussian filter method for post processing of the images captured by STM system or using the add-in provided by the system? What is the advantage compared to other existence methods?

5. Is the deposition control of [7]H molecules on Co/Cu(111) and Fe/W(110) substrate systems reproducible? Authors should provide evidence on this point to strengthen their conclusions.

6. Does the interaction between [7]H molecules and substrates depend on vacuum condition?

Reviewer 3 Report

The authors study the deposition of chiral heptahelicene by sublimation under ultra-high vacuum onto Cu(111), Co(111), and Fe(110) b y low-temperature spin-polarized STM/STS. In general, the paper is of high value for reference and study with clear structure. However, here are some problems to be addressed before its acceptance.

1. Throughout the paper, the main characterization technique seems only STM/STS. Can the authors provide some diffraction results upon samples?

2. In Table I, the authors summarized the [7]H adsorption orientations on Cu(111), Co/Cu(111), and Fe/W(110). Perhaps, a histogram may be clearer and more direct?

3. In Figure A1, the authors mentioned high-resolution topographic image. In my opinion, the term “high-resolution TEM” is common. But how do the authors define a high-resolution topographic image?
